# Emergence of Object Segmentation in Perturbed Generative Models

**Adam Bielski**
University of Bern
adam.bielski@inf.unibe.ch

**Paolo Favaro**
University of Bern
paolo.favaro@inf.unibe.ch

## Abstract

We introduce a framework to learn object segmentation from a collection of images without any manual annotation. We build on the observation that the location of object segments can be perturbed locally relative to a given background without affecting the realism of a scene. First, we train a generative model of a layered scene. The layered representation consists of a background image, a foreground image and the mask of the foreground. A composite image is then obtained by overlaying the masked foreground image onto the background. The generative model is trained in an adversarial fashion against a discriminator, which forces the generative model to produce realistic composite images. To force the generator to learn a representation where the foreground layer corresponds to an object, we perturb the output of the generative model by introducing a random shift of both the foreground image and mask relative to the background. Because the generator is unaware of the shift before computing its output, it must produce layered representations that are realistic for any such random perturbation. Second, we learn to segment an image by defining an autoencoder consisting of an encoder, which we train, and the pre-trained generator as the decoder, which we fix. The encoder maps an image to the input of the generator, which then outputs a composite image matching the original input image. Because the generator outputs an explicit layered representation of the scene, the encoder learns to detect and segment objects. We demonstrate this framework on real images of several object categories.

## 1 Introduction

The problem of extracting a useful interpretation from an image may be simplified by *image segmentation*, i.e., by partitioning such image into regions associated to separate objects. One of the challenges in image segmentation is the difficulty of formulating a precise definition of the "correct image partition". This has been addressed by using manual annotation indicating the pixels associated to a set of object categories (by using bounding boxes or landmarks or detailed object segmentations). However, manual annotation is costly and time-consuming and not easily scalable in some image domains due to lack of expertise and data privacy (e.g., in medicine). This raises the question of whether it is possible to obtain object segmentation directly from collections of real images without any manual annotation. In this paper we show that this is indeed possible and demonstrate it on several datasets of real images. Our approach is to first identify a powerful and general principle to define what an object segment is, and then to devise a model and training scheme to learn through that principle.

With reference to Fig. 1, we propose to build a generative model that outputs a background image, a foreground object and a foreground mask. This model is trained in an adversarial manner against a discriminator. The discriminator aims to distinguish the composite image, obtained by overlaying the output triplet of the generator, from real images. This training alone provides no incentive for the generator to produce triplets with correct object segmentations. In fact, a trivial solution is to have

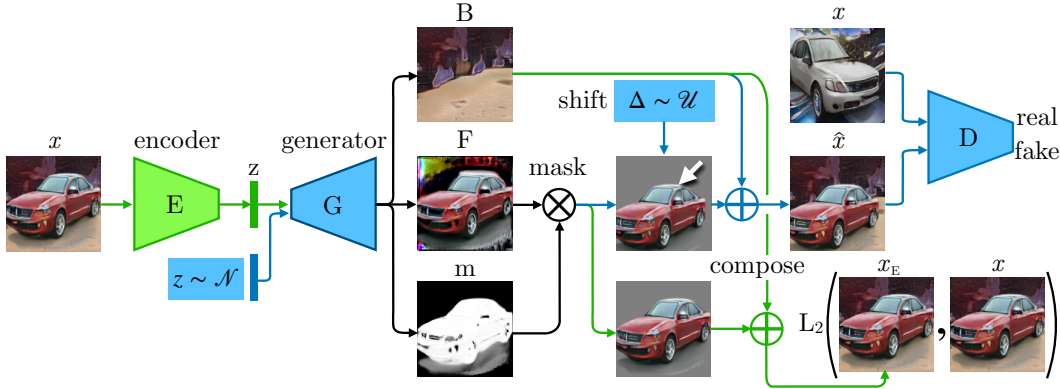

Figure 1: Illustration of the proposed architecture to learn to generate realistic layered scene representations through $G$ (blue path) and to learn to map images to a layered representation through $E$ (and $G$), i.e., to segment objects (green path). The layered representation consists of three components: 1) a background image $B$, 2) a foreground image $F$ and 3) a(n alpha matte) mask image $m$. A crucial component of our model is the generation of random shifts $p$ of the foreground object (in particular, such that they are independent of the input vector $z$ to $G$) during the training of the generator. The generator is trained adversarially against a discriminator $D$. Once the generator $G$ is trained, the encoder $E$ can be trained to extract $z$, which encodes the layered representation.

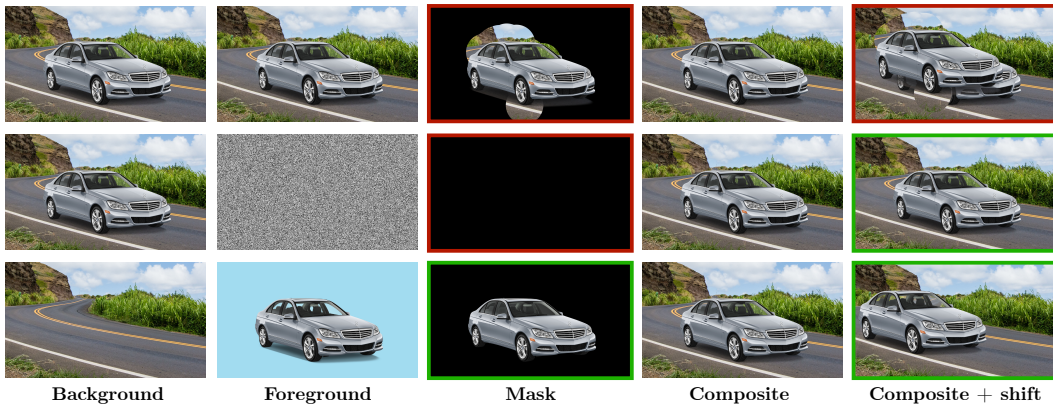

**Background**     **Foreground**     **Mask**     **Composite**     **Composite + shift**

Figure 2: **First row**: Trivial solution, where the background and the foreground are identical and any mask produces a valid composite scene. However, a random foreground shift reveals the invalid segmentation. **Second row**: Trivial solution, where the whole scene is generated in the background and the mask is always empty. **Last row**: The scene after a random shift is valid only when the background generation and the object segmentation are valid and the mask is not empty.

the same realistic image for the foreground and background and a random mask (see Fig. 2, first row). To address this failure, our framework introduces the concept of learning through the perturbation of the model output. According to our object segment definition, we could introduce a small random shift between the foreground and background outputs and still obtain a realistic composite image. If this perturbation is unknown to the generator before producing its triplet, then it is forced to output realistic object segments (see Fig. 2, last row). As a separate step to retrieve the segmentation of an image, we propose to train an encoder network. The encoder is paired with the generator so as to form an autoencoder. The encoder maps an image to a feature vector, which is fed as input to the generator, so that it outputs a triplet that can be used to rebuild the input image. The combination of both of these steps allows us to train the encoder to detect and to segment objects in images without any manual annotation (see the green path in Fig. 1).

**Contributions.** We introduce a fully unsupervised learning approach to segment objects. Unlike in prior work, we do not make use of object detectors, classifiers, bounding boxes, landmarks or pre-

trained networks. To our knowledge, this is the first such solution. Moreover, the proposed method is quite general as we demonstrate it on several object categories qualitatively, and quantitatively on the LSUN Car dataset (Yu et al. [2015]) with Mask R-CNN (He et al. [2017]) used as ground truth as well as CUB-200 dataset with provided annotations. Although we evaluate our approach on a dataset with a single object category at a time, our framework can potentially work on mixed object collections (see Fig. 6). We use single object category datasets because of current GAN limitations.

## 2 Prior Work

Object segmentation is a problem that has been addressed quite successfully via supervised learning given a large dataset of manually annotated images. Mask R-CNN by He et al. [2017] is an example of recent state of the art work that simultaneously detects and segments objects in an image, and can be applied to the segmentation of a wide variety of categories, as shown by Hu et al. [2018]. Since the cost of manually extracting the segmentations in large datasets is very high, a growing effort is devoted to developing methods that minimize the amount of required manual annotation.

In the recent years, several unsupervised and weakly supervised learning methods for segmentation based on deep learning have been proposed (see, for instance, Khoreva et al. [2017], Ji et al. [2018], Hu et al. [2018], Kanezaki [2018], Ostyakov et al. [2018], Remez et al. [2018], Xu et al. [2017], Lutz et al. [2018], Rother et al. [2004], Gupta and Raman [2016], Kim and Park [2015]). These methods require some weak supervision, for example, in the form of initialization, proposals, bounding boxes, or pre-trained boundary detectors.

To avoid manual annotation, one can cast the task of interest in the unsupervised learning framework (see Ghahramani [2004], Barlow [1989]). Early fully unsupervised methods for segmentation relied on a form of clustering of color, brightness, local texture or some feature encoding. For instance, the superpixel clustering method of Achanta et al. [2012] and the mean-shift method of Comaniciu and Meer [2002] are some of the first segmentation approaches based on prescribed low-level statistics. Unsupervised segmentation can be also formulated as a pixel-wise image partitioning task. Ji et al. [2018] define the task as a classification problem with a known number of segment types, such that the mutual information between the predicted partitions of transformed versions of the same image is maximized. The partitioning task is solved by an autoencoding architecture as done by Xia and Kulis [2017]. Rather than using mutual information they constrain the encoding of an image to minimize a normalized cut and some spatial smoothness constraints. This method relies on the distributed representation learned through neural networks. Kanezaki [2018] introduces a method for image segmentation without supervision that relies on three general constraints. The constraints are, however, too generic and thus they cannot always guarantee to yield the correct object segmentation. Several methods for unsupervised scene decomposition were proposed, including Eslami et al. [2016] and Burgess et al. [2019] that use spatial attention and Greff et al. [2016, 2017, 2019] that model images as spatial mixture model to perform unsupervised segmentation. However, these approaches were only shown to work on simpler synthetic or controlled datasets. Other works focus solely on layered image generation. van Steenkiste et al. [2018] propose a GAN that generates a background and individual objects by modeling their relational structure with attention, however the method is shown to work only on simpler datasets. Kwak and Zhang [2016] generates parts of an image with a GAN and RNN and apply restrictions on alpha channels to avoid degenerate solutions, but their blending procedure does not encourage resulting composite images to necessarily contain the exact generated parts. Yang et al. [2017] uses GAN and LSTM to generate a background, a foreground with a mask and a transformation matrix to learn where to place the objects with a spatial transformer. In contrast, in our approach the generator is not aware of the transformations applied to the object.

The work that most closely relates to ours is by Remez et al. [2018]. In this paper the authors build on the idea that realistic segmentation masks would allow the copy-pasting of a segment from one region of an image to another. This remarkable principle can be used to define what an object is. More in general, one could say that pixels belonging to the same object should be more correlated than pixels across objects (including the background as an object). The weak correlation between object and background is what allows introducing a shift without compromising the realism of the scene. However, the weak object-background correlation means also that not all shifts yield plausible scenes. This is why Remez et al. [2018] study the object placement and introduce some randomized heuristics as approximate solutions. In contrast, in our work we avoid heuristics by noticing that small shifts are almost always valid. The price to pay is that background inpainting is required. That is why we

introduce a generative model that learns to output a background and a foreground image in addition to the segmentation mask. One important aspect in the design of unsupervised learning methods is to avoid degenerate solutions. Remez et al. [2018] build a compositional image generator as done by Ostyakov et al. [2018] and then train a segmenter adversarially to a discriminator that classifies images as realistic or not. A degenerate solution for the segmenter is to avoid any segmentation, as the background looks already realistic. The authors describe two ways to avoid this scenario: One is that the dataset of real images contains objects of interest and therefore an empty background would be easily detected by the discriminator. The second is that a classification loss (pre-trained on object identities) would ensure that an object is present in the composite scene. This approach assumes some knowledge about objects (e.g., where they are) and works well on relatively small images ($28 \times 28$ pixels). In contrast, our approach does not require such assumptions and we show its performance on (relatively) high resolution images. In our approach we require that the mask has a minimum number of non zero pixels, i.e., we learn to generate segments with a minimum size (this avoids the degeneracy illustrated in the second row of Fig. 2). This is not a restriction, because we are not making assumptions on single images, but, rather, on the distribution of the image dataset. Then, we establish the correspondence between images and segments in a second step where we train an encoder network. The encoder learns to map images to a suitable noise vector for the pre-trained generator, such that it outputs background, foreground and mask that autoencode the input image after composition (see Fig. 1). The design of such generative models is only possible today thanks to the progress driven by the latest generative adversarial networks of Karras et al. [2018b], which we exploit in this work.

## 3 Learning to Segment without Supervision

Our approach is based on two main building blocks: A generator G and an encoder E (see Fig. 1 for an illustration of the proposed method). The generator is trained against a discriminator in an adversarial manner with the latest high-quality StyleGAN (generative adversarial network) by Karras et al. [2018b,a]. G learns to generate composite scene samples to the extent that the discriminator cannot distinguish them from real images. There are several important aspects that we would like to highlight. Firstly, the training requires no correspondence between the real images and the generated scenes. It allows us to impose constraints on the average type of generated scenes we are interested in, rather than a per-sample constraint. For example, we expect the average scene to have an object with a support of at least $15\% - 25\%$ of the image domain, a condition that may not hold in each sample. Secondly, during training we introduce a random shift unknown to the generator. Thus, the generator must output a background and a foreground that can be combined with arbitrary small relative shifts and still fool the discriminator into believing that the composite image is realistic. This is an implicit way to define what an object is, that avoids manual labelling altogether. The second building block in our approach is an encoder E that learns to segment images. The encoder followed by the generator and the image composition form an autoencoder. The encoder E maps a single image $x$ to a feature vector z that, once fed through the generator, yields its background B (with inpainting), its foreground object F, and its foreground object mask m. The correspondence between images and their object segmentation is thus obtained through the training of the encoder. In the following sections, we explain our approach more in detail.

### 3.1 A Generative Model of Layered Scenes

Consider an $N \times M$ discrete image domain $\Omega \subset \mathbb{Z}^2$. In our notation, we consider only grayscale images for simplicity, but in the implementation we work with color images. We define the representation of a scene as a layered composition of 2 elements: a background image $B : \Omega \mapsto \mathbb{R}$ and a foreground image $F : \Omega \mapsto \mathbb{R}$. Although the foreground is defined everywhere, it is masked with an alpha matte $m : \Omega \mapsto [0, 1]$ in the image composition. The composite image $\bar{x} : \Omega \mapsto \mathbb{R}$ is then defined at each pixel $\mathbf{p} \in \Omega$ as

$$\bar{x}[\mathbf{p}] = (1 - m[\mathbf{p}])B[\mathbf{p}] + m[\mathbf{p}]F[\mathbf{p}]. \tag{1}$$

We define a generator $G : \mathbb{R}^k \mapsto \mathbb{R}^{N \times M}$ through a convolutional neural network (as described in Karras et al. [2018b]) such that, given a $k$-dimensional input vector $z \sim \mathcal{N}(0, I_d)$, it outputs three components $G(z) = [G_B(z), G_F(z), G_m(z)]$, where $G_B(z) = B$, $G_F(z) = F$ and $G_m(z) = m$. The generator is then trained in an adversarial manner against a discriminator neural network

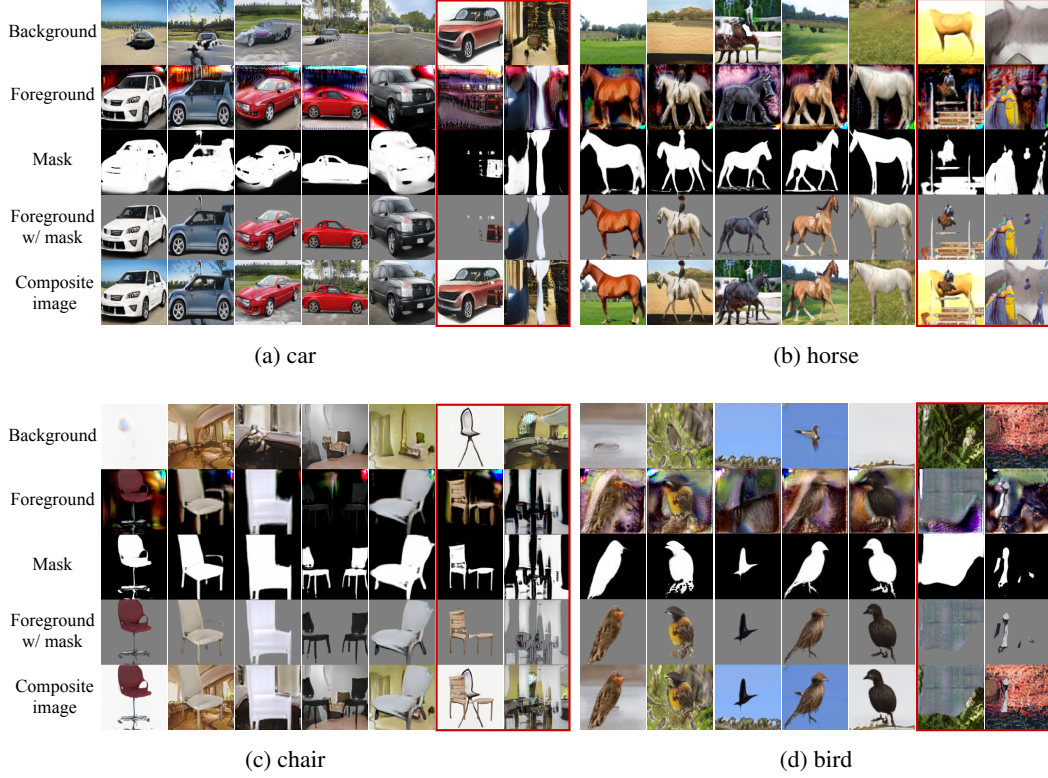

Figure 3: Generated $128 \times 128$ pixels backgrounds, foregrounds, masks, foregrounds with mask applied and composite images for 4 different image categories. Last two columns in each category show generator failures, e.g., an object in the background or an unrealistic foreground.

D $: \mathbb{R}^{M \times N} \mapsto \mathbb{R}$. Our implementation is based on StyleGAN, which, in turn, is based on the Progressive GAN of Karras et al. [2018a], a formulation using the Sliced Wasserstein Distance.

## 3.2  Learning through Model Perturbation

If we trained the generator with fake images according to eq. 1, and we assumed a perfect training, the learned model would be a trivial solution, where the background $G_B(z)$ and the foreground $G_F(z)$ are identical and realistic images, and the mask $G_m(z)$ is arbitrary (see Fig. 2, first row). In fact, there is no incentive for the generator to learn anything more complex than that, and, in particular, to associate foreground objects to the foreground mapping $G_F(z)$. Even the constraint that the average value of the segments $G_m(z)$ should be at least $15\% - 25\%$ is not sufficient to make the generator mapping more meaningful. We use the constraint that foreground objects can be translated by an arbitrary small shift $\Delta \sim \mathcal{U}([-\delta, \delta] \times [-\delta, \delta])$, with $\delta$ a given range of local shifts, and would yield still a realistic composite image. Formally, this can be written by updating eq. 1 as

$$\hat{x}[\mathbf{p}] = (1 - m[\mathbf{p} + \Delta])B[\mathbf{p}] + m[\mathbf{p} + \Delta]F[\mathbf{p} + \Delta]. \qquad (2)$$

Now, the generator has no incentive to generate identical foreground and background images, as a random shift would be immediately detected by the discriminator as unrealistic. Vice versa, it has an incentive to output foreground images and masks that include full objects. If the segments included some background or missed part of the foreground, a small random shift $\Delta$ would also yield an unnatural-looking image that the discriminator can easily detect (in particular, at the segment boundary). Therefore, now the generator has an incentive to output meaningful object segments. To make sure that the mask is non empty, we impose a hinge loss on the average mask value

$$\mathcal{L}_{\text{size}} = \mathbb{E}_{z \sim \mathcal{N}(0, I_d)} \left[ \max \left\{ 0, \eta - \frac{1}{MN} |G_m(z)|_1 \right\} \right] \qquad (3)$$

with a mask size parameter $\eta > 0$ and also use a loss that encourages the binarization of the mask

$$\mathcal{L}_{\text{binary}} = \mathbb{E}_{z \sim \mathcal{N}(0, I_d)} \left[ \min \left\{ G_m(z), 1 - G_m(z) \right\} \right] \qquad (4)$$

Table 1: Ablation study for the LSUN Car dataset. The mean IoU is computed using Mask R-CNN generated segmentations as ground truth. The reference mIoU is computed using masks covering the entire image as segmentation.

| Setting | 64 × 64 pixels | | | 128 × 128 pixels | | |
|---|---|---|---|---|---|---|
| | mIoU | reference mIoU | detected cars | mIoU | reference mIoU | detected cars |
| (a) Default parameters | 0.685 | 0.440 | 6293 | 0.533 | 0.432 | 7090 |
| (b) No shift $\delta = 0$ | 0.039 | 0.428 | 6738 | 0.025 | 0.419 | **7578** |
| (c) 25% shift $\delta = 0.25 \cdot size$ | 0.144 | 0.434 | 6493 | 0.094 | 0.426 | 7259 |
| (d) Bg contrast jitter = (0.7, 1.3) | **0.765** | 0.454 | 6089 | **0.673** | 0.436 | 7046 |
| (e) No random crops | 0.264 | 0.374 | 6339 | 0.136 | 0.365 | 7520 |
| (f) Mask size $\gamma_1 = 10.0$ | 0.733 | 0.443 | 6245 | 0.643 | 0.430 | 7241 |
| (g) Min. mask size $\eta = 5\%$ | 0.693 | 0.458 | 6202 | 0.552 | 0.430 | 7256 |
| (h) Single generator | 0.550 | 0.446 | **6903** | 0.484 | 0.435 | 7544 |

Table 2: Segmentation results. For the LSUN Car dataset, segmentations generated with Mask R-CNN for 10,000 images are used as ground truth; we show the mIoU computed only on images with **detected** cars or using empty masks as ground truth otherwise (**all**). For CUB-200 real ground truth is available. In the last row we use the entire area of an image as a mask.

| Setting | LSUN Car (mIoU) | | CUB-200 |
|---|---|---|---|
| | detected | all | mIoU |
| Our method | 0.540 | 0.479 | 0.380 |
| GrabCut | 0.559 | 0.499 | 0.453 |
| Full mask | 0.402 | 0.357 | 0.132 |

Table 3: FID comparison of our proposed GAN and the single output (SO) GAN.

| Setting | FID | FID |
|---|---|---|
| | 64 × 64 | 128 × 128 |
| SO GAN | 27.807 | 21.665 |
| Our GAN | 31.409 | 30.867 |

Finally, to train the generator we minimize the following loss with respect to $G_B$, $G_F$ and $G_m$

$$\mathcal{L}_{\text{gen}} = -\mathbb{E}_{\hat{x} \sim p_{\hat{x}}}[D(\hat{x})] + \gamma_1 \mathcal{L}_{\text{size}} + \gamma_2 \mathcal{L}_{\text{binary}} \tag{5}$$

with $\gamma_1, \gamma_2 > 0$. To train the discriminator we minimize the following loss with respect to D

$$\mathcal{L}_{\text{disc}} = \mathbb{E}_{\hat{x} \sim p_{\hat{x}}}[D(\hat{x})] - \mathbb{E}_{x \sim p_x}[D(x)] + \lambda \mathbb{E}_{\tilde{x} \sim p_{\tilde{x}}}[(|\nabla_{\tilde{x}} D(\tilde{x})|_2 - 1)^2] + \epsilon \mathbb{E}_{x \sim p_x}[D(x)^2] \tag{6}$$

where $p_x$ is the probability density function of real images $x$, we define $\tilde{x} = \zeta x + (1 - \zeta)\hat{x}$ with random $\zeta \in [0, 1]$, $\lambda > 0$ is the gradient penalty strength and $\epsilon > 0$ prevents the discriminator output from drifting to large values, following Gulrajani et al. [2017] and Karras et al. [2018a].

### 3.3 Object Segmentation via Autoencoding Constraints

Once the generator has been trained, we can learn to associate background, foreground and segments to each image. To do that, we can train an encoder E such that it retrieves, through the generator G, a composite image that matches the original input. The encoder $E : \mathbb{R}^{M \times N} \mapsto \mathbb{R}^k$ maps $x$ to $E(x) \in \mathbb{R}^k$. Let us define $x_E \doteq (1 - G_m(E(x))) \odot G_B(E(x)) + G_m(E(x)) \odot G_F(E(x))$. The loss used to train the encoder E can be written as

$$\mathcal{L}_{\text{auto}} = \mathbb{E}_{x \sim p_x} \left| x_E - x \right|_1 + \mathbb{E}_{x \sim p_x} \left| D_{\text{feat}}(x_E) - D_{\text{feat}}(x) \right|_2^2, \tag{7}$$

where the second term is a perceptual loss that uses features from the trained StyleGAN discriminator.

### 3.4 Implementation

Experimentally, we find that current GAN methods are not yet capable of generating high-quality images from datasets of multiple categories (in a fully unsupervised manner). Thus, we mainly demonstrate our method on datasets with single categories. Given the current progress, we expect GANs to soon address this gap, so that our framework can be applied (as is) to learning the segmentation of multiple object categories with a single training dataset. For all experiments all the

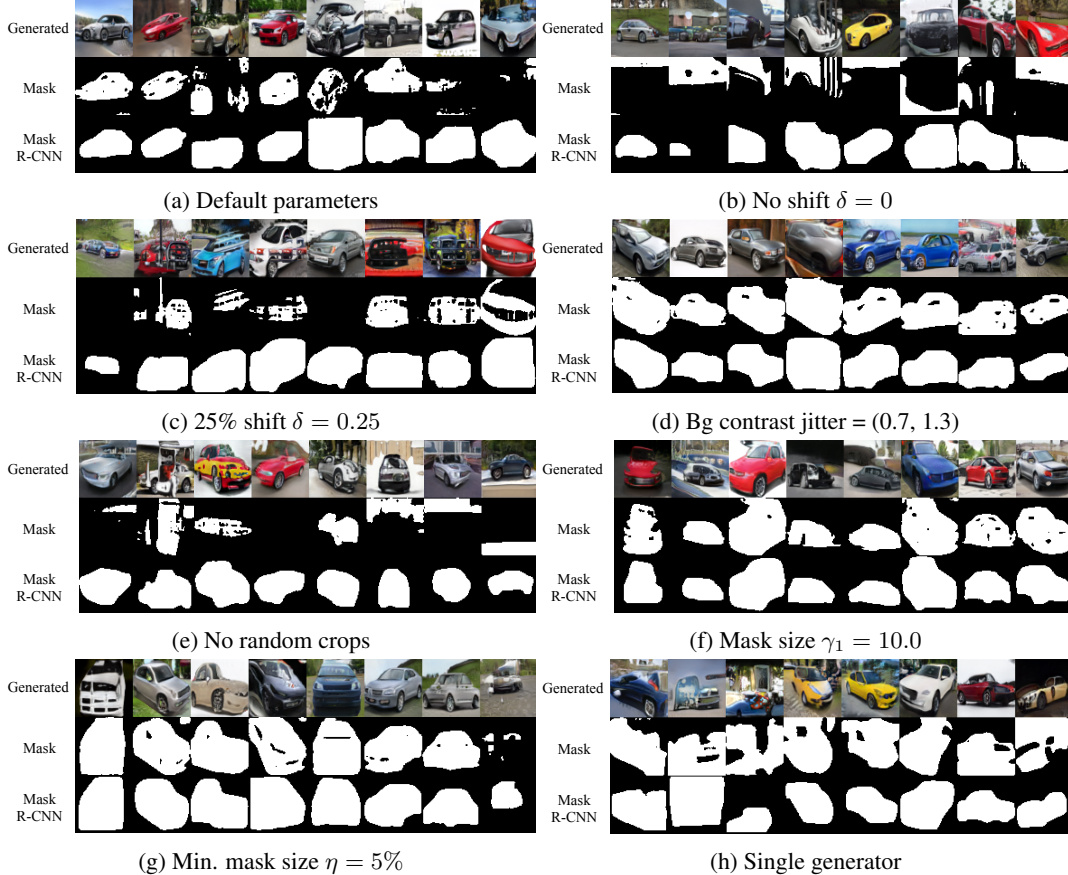

Figure 4: Qualitative results of our approach for settings **(a)-(h)**: generated $64 \times 64$ composite images, masks and outputs of Mask R-CNN.

network architectures and details follow the StyleGAN (Karras et al. [2018b]) if not specified in this section. We use 2 separate generators, one outputs a 3 color channel background, while the other one has two outputs: a 3 color channel foreground and a 1 channel mask followed by a sigmoid activation function. Both generators take the same 512 dimensional Gaussian latent codes as input. We use mixing with probability 0.9 and feed two latent codes to two parts of the generator split by a randomly selected crossover point. We start the training with an initial resolution of $8 \times 8$ pixels and use progressive training to up to $128 \times 128$ pixels. We train with batch sizes 256, 128, 64, 32 and 32 for resolutions $8 \times 8$, $16 \times 16$, $32 \times 32$, $64 \times 64$ and $128 \times 128$ respectively. For each scale the number of iterations is set to process 1,200,000 real images. The local shift range $\delta$ described in section 3.2 is resolution-dependent and set to $\delta = 0.125 \times \texttt{resolution}$. For each resolution of training StyleGAN we first resize the real image to a square image of size $1.125 \times \texttt{resolution}$ and then take a random crop of size $\texttt{resolution}$ to match the shifts in the generated data. We train the StyleGAN network on real images $x$ and composite images $\hat{x}$ (eq. 2) by alternatively minimizing the discriminator loss (eq. 6) and the generator loss (eq. 5). We set the discriminator loss parameters to $\lambda = 10$ and $\epsilon = 0.001$. In the generator loss we set $\gamma_1 = 2$ for the minimum mask size term and $\gamma_2 = 2$ for the binarization term. We optimize our GAN with the Adam optimizer (Kingma and Ba [2015]) and parameters $\beta_1 = 0$, $\beta_2 = 0.99$. We use a fixed learning rate of 0.001 for all scales except for $128 \times 128$ pixels, where we use 0.0015.

## 4 Experiments

We train our generative model on 4 LSUN object categories (Yu et al. [2015]): `car`, `horse`, `chair`, `bird`. For each dataset we use the first 100,000 images. Objects in the datasets show large variability

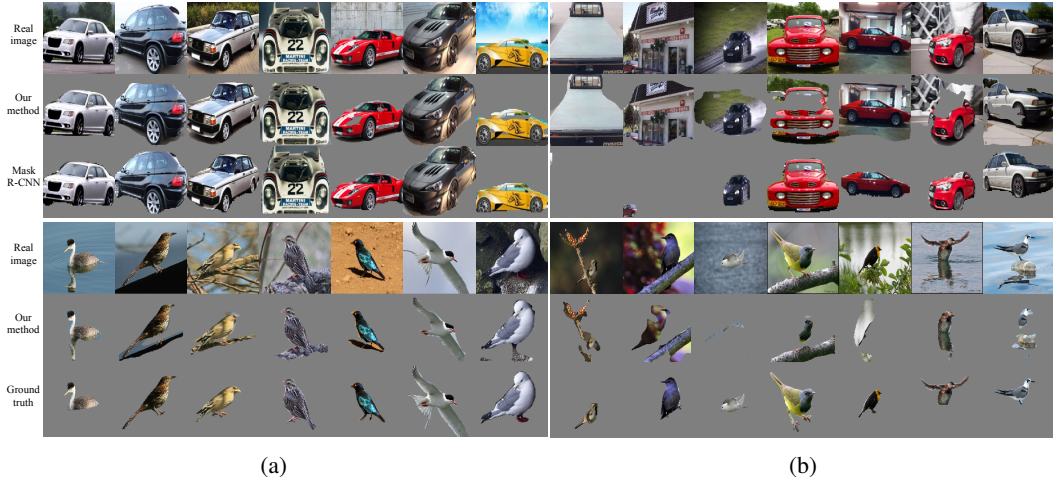

Figure 5: Qualitative results of segmentation on LSUN Car and CUB-200 datasets. (a) Examples of successful segmentations. (b) Examples of failures.

in position, scale and pose. We set the minimum mask size $\eta$ in eq. 3 to 25%, 20%, 15% and 15% for `car`, `horse`, `chair`, `bird` datasets respectively. In Fig. 3 we show some examples of outputs produced by the generators from random samples in the Gaussian latent space. From the first to the fifth row in each quadrant: generated background layer, generated foreground layer, generated foreground mask layer, product between the mask and the foreground layer, and final composite image. As can be seen, the generator is able to learn very accurate object segmentations and texture. Also, it can be observed that the background has some residual artifacts in the center. This is due to the limited shift perturbation range, which does not allow the background layer to receive much feedback from the loss function during the training. In some cases the exact separation between object and background is not successful. This can be seen in the last two columns for each dataset.

**Ablation study.** To validate the design choices in our approach we perform ablation experiments on the LSUN Car dataset. We introduce the following changes to **(a)** the default parameters described in section 3.4, **(b)** disable the shift by setting the range of random location shift $\delta = 0$, **(c)** increase the shift to $\delta = 0.25 \times \texttt{resolution}$, **(d)** randomly jitter the background contrast in the range (0.7, 1.3) to further prevent the background from filling parts of objects, **(e)** directly resize real images to the desired resolution without random cropping, **(f)** increase the strength of the mask size loss $\mathcal{L}_{\text{size}}$ by setting its coefficient $\gamma_1 = 10$, **(g)** set the minimum mask size parameter to a smaller value $\eta = 5\%$, **(h)** use a single generator with 3 outputs for background, foreground and mask. To evaluate the quality of the generated segments, we generate 10,000 images and masks for each setting. We binarize our masks with a 0.5 threshold. To obtain an approximated mask ground truth on generated composite images we run Mask R-CNN (He et al. [2017], Massa and Girshick [2018]) pre-trained on MS-COCO (Lin et al. [2014]) with a ResNet50 Feature Pyramid Network backend. If the car is detected, we evaluate the mean Intersection over Union (mIoU) with the mask generated by our models on these images. We run the evaluation on $64 \times 64$ and $128 \times 128$ pixels resolution, but resize the $64 \times 64$ images to $128 \times 128$ before feeding them to Mask R-CNN. The quantitative results can be found in Table 1 and the qualitative results in Fig. 4.

Our ablation shows that the random shifts (see section 3.2) are essential in our approach. When not used **(b)** the object segmentation fails and the objects are often in the background. The quality of the segmentation decreases drastically when the random shift range does not correspond to foreground object shifts in real images. This is illustrated by the setting **(c)** with large random shifts and by the setting **(e)**, where the real images are not randomly cropped. The additional random contrast jitter of the generated background **(d)** helps separate the foreground object from the background. A smaller value of the $\eta$ parameter to ensure the minimum mask size **(g)** does not have a big impact on the results: It helps to avoid the empty masks, but the mask size is mainly determined by the realism requirement. Using a single generator **(h)** to produce all outputs makes the background and the foreground too correlated, which prevents it from learning a good layered representation.

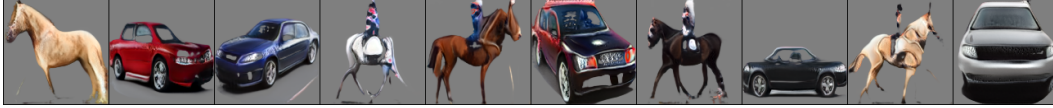

Figure 6: Generated object segments using a dataset with two object categories: cars and horses.

**Quality of generated images.** To evaluate the quality of generated composite images, we compute the Frèchet Inception Distance (FID) (Heusel et al. [2017]) using 10K real and 10K generated images composite images from our model (**d**). We compare it with a standard StyleGAN producing the entire images at once, trained for the same number of iterations. The results are presented in Table 3. The difference in the FIDs may be explained by the more demanding constraints of our model, which may hinder the GAN training.

**Dataset with more than one object category.** Although we run our experiments on datasets containing objects of one category, we argue that our method should work with multiple object categories when the GANs improve and are able to produce realistic images on diverse datasets. To verify this, we train our model on a dataset consisting of 50K images from LSUN Car and 50K images from LSUN Horse datasets. The qualitative results are presented in Fig. 6. Although the quality of generated images on such a dataset is lower, our model is still able to generate segmented scenes.

**Segmenting real images.** Finally, we train an encoder to find the segmentation of real images, as described in section 3.3. We use our best generator trained on LSUN Car $64 \times 64$ images with background contrast jitter (setting (**d**)) and freeze its weights. We train an encoder that produces $5 \cdot 2 \cdot 2 = 20$ latent codes of $512$ dimensions: For each of the 5 StyleGAN scales we get 2 separate codes for AdaIN (adaptive instance normalization) layers in a convolutional block and get separate codes for 2 generators (background and foreground with mask). For the encoder we use a randomly initialized ResNet18 network (He et al. [2016]) with a $64 \times 64$ input without average pooling at the end and add a fully-connected layer with a $512 \cdot 20 = 10240$ output size. We feed the codes to the generator and minimize the autoencoder loss (eq. 7). In the perceptual loss we use our discriminator to extract $512 \times 8 \times 8$ spatial features on real and generated images. We evaluate the segmentation on the first 10,000 images of the LSUN Car dataset. We train separate encoders on chunks of 100 images as we found that it makes the encoding more stable than training on the entire dataset. We run the training for 1000 iterations with Adam optimizer, learning rate of $0.0001$ and $\beta_1 = 0.9$, $\beta_2 = 0.999$. After training, we encode the images and feed the codes to the generator to obtain the masks. For the approximated ground truth we run Mask R-CNN on real images and evaluate our segmentation with mean IoU. We also compute the mIoU using the output of the GrabCut algorithm and a naive mask covering the entire image. The results are presented in Table 2. The performance of our method is capped by ambiguities in inverting the generator with an encoder, which is an active topic of research. We present sample segmentation results in Fig. 5. We notice some failures especially in the case of small objects. We repeat the same training and evaluation procedure on Caltech-UCSD Birds-200-2011 dataset (Wah et al. [2011]), for which the segmentation ground truth is available. We use the parameters that worked best on the LSUN Car dataset for training both the generator and the encoders.

## 5 Conclusions

We have introduced a new framework to learn object segmentation without using manual annotation. The method is based on the principle that valid object segments can be locally shifted relative to their background and still yield a realistic image. The proposed solution is based on first training a generative network to learn an image decomposition model for a dataset of images and then on training an encoder network to assign a specific image decomposition model to each image. It strongly relies on the accuracy of the generative model, which today can be built with adversarial techniques. However, it is a quite general framework that can be easily extended. For example, the current generative model postulates that a scene is composed only of a foreground and a background object, but a simple increase of output layers would allow describing scenes with multiple objects.

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
