[Reviews · NeurIPS 2019]

Reviewer 1



Post-rebuttal: Thank you to the authors for submitting a thoughtful rebuttal. The other reviews didn't raise any new concerns that would sway my original rating. Originality: The main originality of this work is the observation that small perturbations in objects result in valid scenes. This allows the method to solve a much broader problem than Remez 2018, without the need for object detectors and heuristics for where to position the pastes. While Remez is limited to generating object masks for object detectors (e.g. weakly supervised instance segmentation), this work is able to also consider Quality, Clarity: Overall, I think think the writing is clear and the experiments are thorough. Evaluation: The authors do a good job of validating the contributions with ablation studies and show that indeed this foreground shifting is essential to the performance of the technique. I think evaluating with Mask R-CNN is a clever idea for a dataset without annotations, but it raises a greater concern: Why not just train this method on a dataset with a labeled test set? Does this method require very object-centric images? Would this method work for MS COCO? Does it matter that it doesn't work for MS COCO? Further, how would this work compare to Remez in their experimental setting - i.e. how would your method work for learning to generate object masks from detections? It seems natural to do this experiment given the similarity to Remez. One important comment: This method is not fully unsupervised (line 62) -- a huge amount of effort goes into collecting a dataset that only contains the category of interest. I think it's important to clarify this..

Reviewer 2



This paper presents a generative model of layered object representation where a generator synthesizes a foreground object, a foreground mask and a background to compose an image at the same time while a discriminator tells the composite's realism. To prevent the generator from cheating, i.e. generating the same foreground and background with a random mask, this paper proposes to add random shift to the generated foreground object and its mask. The rationale behind this is that the generated foreground object and its mask must be valid to allow such random shifts while maintaining the realism of the scene. In other words, the foreground can move independently on the background in a valid layered scene. As a result, this paper discovered that a high-quality foreground mask emerges from this layered generative model so that an encoder is trained to predict the mask from an image for object segmentation. --Originality A series of layered generative models have been proposed such as Attend-Infer-Repeat (AIR) (Eslami et al., NIPS 2016) and Generative Modeling of Infinite Occluded Objects (Yuan, et al. ICML 2019) but they can only handle synthetic scenes. The "Attribute2Image" model (Yan et al. ECCV 2016) proposed a similar layered generative model but its mask is supervised. To my knowledge, this paper is the first layered generative model that can synthesize real images without mask supervision. A thorough survey on this topic would improve the originality of this paper. This paper discussed some of weakly-supervised/unsupervised object segmentation models but it is not clear how well it compares with. --Quality This paper clearly states the basic assumption of 15%-25% support of an object and formulates a corresponding loss to it (Equation 3). The proposed generative model is trained based on state-of-the-art GAN algorithms and yields high-quality samples on a few object categories. The generated masks in Figure 2 have impressive details. A detailed evaluation of image generation quality, especially comparing with the generative models without layered representations would be very helpful to improve this paper's quality. This paper did a fairly good ablation study on different parameters, but it is not clear to me why it uses the segmentation masks computed from Mask RCNN as ground truth. Why not use the actual ground truth and compare with those supervised results? I believe \eta in Equation 3 is a pretty important parameter that decides how many foreground pixels the model is expected to generate. It would be useful to perform an analysis by varying its value. I'm curious if this paper has considered other priors, e.g. mask connectivity, for unsupervised segmentation. This paper is closely related to the work by Remez et al. in ECCV 2018 as stated, so an experimental comparison with it and other baselines, e.g. grab cut would be useful to improve the paper quality. --Clarity Overall, the clarity is good. As this paper uses 2 generators for foreground and background separately, it will be good to make it clear in Figure 1. --Significance This paper presents a nice step forwards towards unsupervised learning of layered generative models, which have been shown to have the potential for high-quality image synthesis and scene understanding. Unsupervised segmentation is also an interesting problem in computer vision towards reducing human annotations/labors. Overall, I like this paper but also believe it has large room to improve in terms of literature review, experiment setup and evaluation. ======After rebuttal======= I'm mostly satisfied with the rebuttal and request the authors to add literature and baselines as promised.

Reviewer 3



This paper introduces a simple but highly original idea, leading to a really impressive result. The basic idea is similar to Remez '18 "Learning to Segment via Cut-and-Paste", which also uses a GAN to generate a mask in an unsupervised fashion, and a discriminator to distinguish composite images from natural ones. However this paper's use of a three-part generative model, producing a background, foreground, and a mask simultaneously, completely avoids the problem(s) that arise in Remez around trying to identify where the object can be plausibly pasted into a pre-existing background (i.e. must not overlap the original object, placement must be realistic - no cars in the sky, resolutions of background and foreground images must be the same, etc.), as well as the failure mode where the mask only selects part of the object. Thus I believe the paper has considerable novelty. The paper represents a significant step towards learning to segment objects from images without any ground truth mask data. One drawback is that it does require photo-realistic image generation, which currently can only be accomplished for a limited set of classes -- and usually only one class at a time -- by state-of-the-art GAN models. Currently this is somewhat limiting as a segmentation model, but the problem setup feels like a novel image (background + object + mask) generation problem. As such, I could see this having significant impact in both the image generation and segmentation communities. The paper is well written, clear and easy to follow, with some very minor exceptions that I've called out in the "Improvements" section below. The experiments are sufficiently comprehensive, although it would be interesting to see in more detail how the approach performs on images with partially-occluded foreground objects, and to how "central" the foreground object must be. When presented with two foreground objects, does mask generation tend towards a semantic mask (the chairs in Fig 2c.), or does it produce an instance mask for only one of the instances? Overall a very nice paper. I enjoyed reading it. ==== UPDATE: I have reviewed the insightful comments from the other reviewers, as well as the authors' response. I believe it addresses all of the major concerns. An expanded literature review using Reviewer #2's suggestions would be helpful, but the omissions are not fatal to this paper. Based on this I have decided to leave my score as-is.

[Author Response · NeurIPS 2019]

We thank all the reviewers for their valuable feedback. We hope that our answers below clarify all concerns.

**Evaluation on a labeled dataset.** In the paper we used LSUN datasets because it provides many examples of images
for a single category and this is important for the quality of GAN. Since this dataset is not annotated, we used Mask
R-CNN to get approximated ground truth segmentation on LSUN Car dataset. To address concerns of ***reviewer 1*** and
***reviewer 2***, we trained a model on the annotated Caltech-UCSD Birds-200-2011 dataset (*Wah et al., The Caltech-UCSD*
*Birds-200-2011 Dataset, California Institute of Technology, 2011*). We used the parameters that worked best on car
dataset in ablation studies. We then trained encoders in similar fashion as we did for LSUN Car and ran the evaluation,
obtaining **mean IoU = 0.380**, while the reference box IoU (for the mask covering the entire image) is **0.132**.

**Object-centric datasets / robustness.** We show failure cases in Fig. 4 (b). Our experiments on LSUN datasets show
that the training of our generator is sufficiently robust to foreground objects appearing at different sizes, viewpoints,
locations and where several images have object parts only. Also, see the experiment below with images from two
categories. However, because of current GAN limitations, we may not be able to capture all the data distribution modes
(small scales, extreme shifts, etc.).

**Dataset with more than 1 object category.** *Reviewer 1* noticed that a huge amount of effort goes into collecting a
dataset that only contains the specific category of interest. We argue that our method should work with multiple object
categories when the GANs improve and are able to produce realistic images on diverse datasets. To show this, we
trained our model on a dataset consisting of 50k images from LSUN Car and 50k images from LSUN Horse images.
Although the quality of generated images on such a dataset is lower, our model is able to generate segmented scenes for
this dataset consisting of two object categories, as presented in the image below.

However, we could not work on the MS COCO dataset because current GANs are not able to deal with its complexity.

**Importance of $\eta$ parameter.** *Reviewer 2* expressed concerns about the importance of the $\eta$ parameter, which defines
the minimum area of the foreground object. We ran an additional ablation experiment with $\eta = 5\%$ and found that
the results are similar to our default parameters with $\eta = 25\%$, as presented in the table below. The loss term for the
minimum area of the foreground object helps avoid the degenerate solution (empty masks), but the generated objects
must look real and this is what determines their size.

| Setting | $64 \times 64$ | | | $128 \times 128$ | | |
|---|---|---|---|---|---|---|
| | mIoU | reference mIoU | detected cars | mIoU | reference mIoU | detected cars |
| Default parameters $\eta = 25\%$ | 0.685 | 0.440 | 6293 | 0.533 | 0.432 | 7090 |
| $\eta = 5\%$ | 0.693 | 0.458 | 6202 | 0.552 | 0.43 | 7256 |

**Evaluation of image generation quality.** As *reviewer 2* suggested, we ran an evaluation of the quality of images
generated by our model. We compute FID using 10k real and 10k generated composite images from our model and
compare it with a standard StyleGAN producing the entire images at once, trained for the same number of iterations.
The results are presented in the table below. The difference in the FIDs may be explained by the more demanding
constraints of our model, which may hinder the GAN training.

| Setting | **FID** $64 \times 64$ | **FID** $128 \times 128$ |
|---|---|---|
| Single output GAN | 27.807 | 21.665 |
| Our GAN | 31.409 | 30.867 |

**Prevention from all-ones masks.** *Reviewer 4* mentioned the danger of getting a degenerate solution in which all of
the masks are ones. In our approach random shifts of the foreground object prevent this solution. If the mask consists of
all ones, a shift will create a detectable transition between the exposed background and the foreground.

**Training the encoder.** *Reviewer 4:* To obtain the segmentations, we split the dataset into sets of 100 images and train
a separate encoder for each set. We then use the encoder to extract the latent codes for its set. We found the training to
be more challenging as we added more images.

**Literature review.** While we focused on prior work on unsupervised and weakly supervised segmentation, *reviewer 2*
suggested including more works on layered generative models. We will add those and others to better underline the
contribution and novelty of our method, which works on real images and without supervision.

**Comparison with cut-and-paste, other baselines.** We could not prepare the comparison in time for the rebuttal but
we will add it in the camera-ready paper if accepted.

[Meta-Review · NeurIPS 2019]

The paper presents a scheme for learning object segmentation from a set of image data without annotation. The main assumption upon which the approach is built is that the location of object segments can be perturbed relative to background. The method is shown to empirically improve segmentation performance on real images of several object categories. All reviewers have found the contributions of this work significant and interesting, both in terms of the methodology (simple but original) and empirical results. Please consider the improvements suggested by reviewers in the final version.